# ATP increases murine neuroblastoma cell size through a PANX1- and macropinocytosis-dependent mechanism

Andrew K. J. Boyce[1,2,*], Haifei You[1], Leigh E. Wicki-Stordeur[1] and Leigh Anne Swayne[1,*]

## ABSTRACT

Macropinocytosis is an endocytic process that allows cells to respond to changes in their environment by internalizing nutrients and cell surface proteins, as well as modulating cell size. Here, we identify that adenosine triphosphate (ATP) triggers macropinocytosis in murine Neuro2a neuroblastoma cells, driving an increase in cell size, and internalizing the ATP release channel pannexin 1 (PANX1) to macropinosomes. Amiloride treatment and mutation of an extracellular tryptophan (W74) in PANX1 abolished ATP-evoked cell area enlargement, suggesting that PANX1 may itself regulate this form of macropinocytosis. Transient expression of the GTP-hydrolysis resistant ADP-ribosylation factor 6 GTPase (ARF6 Q67L) led to increased cell size, PANX1 internalization and localization to endosomal compartments, consistent with macropinocytosis. Inhibiting macropinocytosis-associated GTPases, phosphoinositide-3 kinase (PI3K), and disrupting actin polymerization abolished ATP-induced PANX1 internalization, supporting a macropinocytic mechanism. Further, these inhibitors disrupted co-distribution of intracellular PANX1 with macropinosomal cargo. Several lipid-PANX1 interactions were identified with relevance to macropinocytic mechanisms. The role of PANX1 in ATP-mediated macropinocytosis could be particularly important for disease states implicating PANX1, such as cancer, where ATP can act as a purinergic regulator of cell growth/metastasis and as a supplementary energy source following internalization.

KEY WORDS: Pannexin, PANX1, Macropinocytosis, Endocytosis, Internalization, Purinergic signalling, ATP, Adenosine triphosphate

## INTRODUCTION

Amongst the many ways that cells adapt to environmental changes, several key modifications include cell surface receptor density, nutrient acquisition, and cellular structure. Macropinocytosis, or 'cell drinking', is a non-canonical membrane internalization process that enables these tightly associated dynamic adaptations (Swanson and King, 2019). In some cell types, macropinocytosis enables recycling of the equivalent of the full cell surface area once every 30 min (Steinman et al., 1976). A more thorough appreciation of the regulation of macropinocytosis is important for understanding

antigen presentation in phagocytic immune cells (Roche and Furuta, 2015) and receptor-mediated signalling localized to endosomal compartments (Yoshida et al., 2015; Wall et al., 2017), as well as many types of viral and pathogenic bacterial entry (de Carvalho et al., 2015; Saeed et al., 2010; Mercer and Helenius, 2012; Francis et al., 1993) and nutrient acquisition in tumorigenic cells (for reviews, see Finicle et al., 2018 and Bloomfield and Kay, 2016). In the context of cancer, macropinocytic mechanisms are hijacked by tumour cells for internalization of adenosine triphosphate (ATP) to promote cell growth and survival (Wang et al., 2017; Qian et al., 2014). Here, extracellular ATP provides energy status information in the tumour microenvironment (Shukla et al., 2024) by way of purinergic signalling through ATP-sensitive purinergic receptors and ATP release mechanisms, including the large pore channel pannexin 1 (PANX1) (Laird and Penuela, 2021).

In our previous studies, we identified that PANX1 is internalized in response to elevated extracellular ATP via an undetermined non-canonical endocytic mechanism in murine neuroblastoma cells (Boyce and Swayne, 2017; Boyce et al., 2015). PANX1 is widely understood to promote tumour cell survival in various cancers (Laird and Penuela, 2021), including the proliferation and differentiation of murine neuroblastoma (Wicki-Stordeur and Swayne, 2013; Wicki-Stordeur et al., 2012) and malignancy of human neuroblastoma (Langlois et al., 2023). In this study, we investigate the hypothesis that ATP-mediated PANX1 internalization occurs through macropinocytosis and drives a PANX1-dependent increase in cell size. Macropinocytosis is distinct from classical types of membrane internalization or endocytosis in that it does not rely on coat and scission proteins used in clathrin- and caveolin-mediated endocytosis. Instead, during macropinocytosis, discrete regions of cholesterol and phosphatidylinositol-4,5-bisphosphate [PI(4,5)P$_2$]-rich plasma membrane and its associated receptors are engulfed via actin-dense membrane ruffles (Grimmer et al., 2002). Initially, ADP-ribosylation factor 6 (ARF6), a small GTPase resident to the membrane ruffles, activates phosphatidyl inositol-4-phosphate 5-kinase (PI4P5K) (Naslavsky et al., 2003) to generate PI(4,5)P$_2$ (Honda et al., 1999; Brown et al., 2001). Generation of PI(4,5)P$_2$ enables recruitment of the small GTPases Ras-related C3 botulinum toxin substrate 1 (Rac1) and cell division control protein 41 (more commonly known as CDC42) to the ruffling membrane (Zhang et al., 1999; Daste et al., 2017), which then activate p21-activated kinase (Dharmawardhane et al., 2000). Next, p21-activated kinase phosphorylates a protein called brefeldin A-dependent ADP ribosylation substrate to initiate membrane curvature (Liberali et al., 2008), which is enabled by conversion of lysophosphatidic acid to phosphatidic acid (for review, see Bohdanowicz and Grinstein, 2013), leading to the formation of large vesicles ranging in diameter from 0.2 to 5 µm, referred to as macropinosomes. Given the lack of coat protein, structural specificity, or unique membrane-bound molecules, macropinosomes are difficult to distinguish from other endocytic compartments of similar size, and consequently, are commonly identified using fluorescently labelled

[1]School of Medical Sciences, Faculty of Health, University of Victoria, Victoria, BC V8P 5C2, Canada. [2]Department of Neurosciences, University of New Mexico School of Medicine, Albuquerque, NM 87131, USA.

*Authors for correspondence (anboyce@salud.unm.edu; lswayne@uvic.ca)

A.K.J.B., 0000-0002-3708-9884; L.A.S., 0000-0001-5852-3144

fluid-phase endocytosis markers (i.e. FITC-dextrans; Lin et al., 2020). Fortunately, the small GTPases that regulate macropinocytosis are uniquely sensitive to changes in sub-membranous pH; and for this reason, amiloride inhibition of the $Na^+/H^+$ exchanger selectively targets macropinocytosis over other forms of endocytosis (Koivusalo et al., 2010).

Depending on the cell type, macropinocytosis can be a constitutive process or triggered by extracellular stimuli and metabolites (Swanson, 2008). Examples of cells that undergo constitutive macropinocytosis include immature dendritic cells (for sampling of soluble antigens; Canton, 2018) and cancerous fibroblasts transformed with oncogenic K-Ras or v-Src (Amyere et al., 2000). Extracellular molecules that can stimulate macropinocytosis include growth factors [e.g. epidermal growth factor (Bryant et al., 2007; Lee et al., 2019; Balaji et al., 2012; Solis et al., 2012; Koumakpayi et al., 2011), macrophage colony-stimulating factor (Racoosin and Swanson, 1989), vascular endothelial growth factor (Basagiannis et al., 2016), platelet-derived growth factor (Schmees et al., 2012) and chemokines (Tanaka et al., 2012)]; this breadth of stimuli enables context-dependent regulation of signalling at the cell surface. Macropinocytosis can also be hijacked by viruses (Bannach et al., 2020; Wang et al., 2015; Sánchez et al., 2012) for cellular entry. Although not a canonical growth factor, ATP plays a growth factor-like role in regulating many cellular functions through activation of purinergic receptors (Wicki-Stordeur et al., 2012; Burnstock and Ulrich, 2011; Franke and Illes, 2006; Zimmermann, 2016; Erlinge, 1998) and can trigger macropinocytosis in human lung cancer cells (Qian et al., 2014), driving an increase in intracellular ATP and promoting drug resistance. In addition to ATP, other macropinosome cargo can include amino acids and other metabolites important for cellular function (i.e. growth, survival, etc.; Yoshida et al., 2015, 2018).

As a part of autocrine and paracrine purinergic signalling, ATP is released via a number of mechanisms, including through channels like PANX1 as well as by vesicular release (Lazarowski, 2012). PANX1 forms ubiquitously-expressed heptameric channels that are permeable to small ($Cl^-$) and large (e.g. ATP) anions (Ruan et al., 2020; Deng et al., 2020; Mou et al., 2020; Chiu et al., 2017; Michalski et al., 2020; Ma et al., 2009, 2012; Nielsen et al., 2020). Additionally, PANX1 may also regulate the flux of small cations (e.g. $Ca^{2+}$; Nielsen et al., 2024; Yang et al., 2020) and release of large cations (spermidine; Medina et al., 2020). Selectivity for anions or cations has been proposed to depend on the mechanism of channel activation (Sanchez Arias et al., 2020a). PANX1 anion selectivity, characteristic of certain open states (Ruan et al., 2020; Ma et al., 2012; Nielsen et al., 2020), has been attributed to extracellular tryptophan (W74) and arginine (R75) residues that form a molecular filter for size and charge, respectively. While their overall properties are still the focus of intense investigation, it has been well established that PANX1 channels play a role in ATP release (Chiu et al., 2017; Sandilos et al., 2012; Bao et al., 2004; Chekeni et al., 2010; Seminario-Vidal et al., 2011). Once in the extracellular space, ATP and its metabolites, arising from the activity of tissue-specific ectonucleotidases, trigger an array of downstream signalling pathways that can lead to myriad cellular changes (Giuliani et al., 2019).

We and others have demonstrated that extracellular ATP can also directly impact the ATP release machinery (Boyce and Swayne, 2017; Boyce et al., 2015; Ma et al., 2009; Qiu and Dahl, 2009), inducing sustained changes in cell signalling. Exogenously added extracellular ATP can inhibit PANX1 channel activity (Ma et al., 2009; Qiu and Dahl, 2009; Qiu et al., 2012) as well as stimulate PANX1 internalization (Boyce and Swayne, 2017; Boyce et al., 2015). Cell surface physical association of PANX1 with the ionotropic purinergic P2X7 receptor (P2X7R) preceded internalization (Boyce and Swayne, 2017). Site-directed alanine substitution of the PANX1 extracellular loop tryptophan, W74, disrupted cell surface P2X7R-PANX1 association (Boyce and Swayne, 2017), as well as PANX1 internalization (Boyce et al., 2015), suggesting that this residue regulates both ionic selectivity and cell surface stability. In the course of this work, we noted that a considerable amount of internalized PANX1 localized to structures much larger than canonical endocytic vesicles. ATP-triggered internalization of PANX1 did not utilize the canonical clathrin-, caveolin-, and dynamin-associated machinery (Boyce et al., 2015), consistent with work from other groups (Gehi et al., 2011; Bhalla-Gehi et al., 2010). Furthermore, ATP-evoked PANX1 internalization in N2a cells coincided with active filopodial dynamics, relied on cholesterol, and occurred in the absence of canonical endocytic effectors (Boyce et al., 2015), suggesting a non-canonical endocytic process, such as macropinocytosis.

We first identified that ATP triggered an increase in cell size via a mechanism that was disrupted by the internalization-deficient PANX1 W74A mutant and amiloride, consistent with expected cell size changes following macropinocytosis (Lloyd, 2013). Next, we examined the impact of the macropinocytosis blocker amiloride on ATP-induced PANX1 internalization. As anticipated, amiloride, prevented ATP-evoked PANX1 internalization. Expression of a constitutively-active ARF6 GTPase increased cell size and basal intracellular PANX1 levels and prevented further ATP-induced PANX1 internalization suggesting that the macropinocytosis effector plays a role in ATP-induced PANX1 internalization. In the presence of several macropinocytosis inhibitors, intracellular PANX1 did not increase when cells were stimulated with ATP. Further, these inhibitors disrupted co-distribution of PANX1 with the macropinosome cargo, 70 kDa dextran, to intracellular compartments consistent with the size of macropinosomes. Cryo-electron microscopy (cryo-EM) structures have highlighted putative PANX1-lipid interactions (Ruan et al., 2020; Kuzuya et al., 2022), while several recent studies have even suggested that lipid subtypes can permeate the channel (Anderson et al., 2024 preprint; Bialecki et al., 2020) and regulate gating (Kuzuya et al., 2022; Henze et al., 2025). Here, we identify several lipid interactors for the PANX1 C-terminus related to macropinocytosis and internalization mechanisms. Taken together, these results suggest that, in addition to undergoing macropinocytosis in response to extracellular ATP, PANX1 could also play a broader role in the regulation of ATP-regulated macropinocytosis to promote expansion of neuroblastoma cell size.

## RESULTS

### Elevated extracellular ATP increases N2a cell size through a putative PANX1- and macropinocytosis-dependent mechanism

Growth factor-stimulated macropinocytosis is known to increase cell size [i.e. nerve growth factor (Purves et al., 1988), insulin-like growth factor (Laplante and Sabatini, 2012); for review, see Lloyd, 2013]. In our previous study, we identified that ATP-evoked PANX1 internalization could be disrupted by alanine substitution of the extracellular residue W74A. Here, we asked whether PANX1 internalization by ATP stimulation was accompanied by a change in cell size. Our model involved generating stable expression of PANX1-EGFP in murine neuroblastoma N2a cells. This clonal colony was selected for low level expression that distributes both to the cell surface and intracellular compartments (Boyce et al., 2015); however, it is important to highlight that, given ectopic expression in a system with endogenous PANX1, overexpression is expected.

EGFP-tagged PANX1, has a similar subcellular distribution to endogenous PANX1 (i.e. cell surface with some intracellular localization; Penuela et al., 2008). Here, to visualize internalization [using a protocol established in our previous studies (Boyce and Swayne, 2017; Boyce et al., 2015)], we bath applied ATP onto cultured N2a cells in the presence of transient protein-synthesis inhibition, allowing us to visualize steady-state changes in PANX1 localization in cells devoid of *de novo* synthesized protein in the secretory pathway.

To quantify changes in cell size, we acquired confocal Z-stacks and measured the cross-sectional area of vehicle and ATP-stimulated PANX1-EGFP- or PANX1-W74A-EGFP-expressing N2a cells at the plane where the nucleus was centred. Here, a transmitted light image was acquired to identify PANX1-EGFP at the cell periphery (Fig. 1A). The cell periphery was traced and the cross-sectional area was quantified. Elevated extracellular ATP expanded cross-sectional cell area (Fig. 1B), while no change was observed in response to elevated extracellular ATP with PANX1 W74A expression (Fig. 1B), suggesting that PANX1 itself might regulate macropinocytosis.

Given the effect of PANX1 mutation on cell size, we next addressed whether this ATP-evoked increase in cell size was triggered by macropinocytosis and whether PANX1 was internalized in the process. Macropinocytosis involves the coordinated recruitment and activation of several pH-sensitive GTPases in membrane ruffles (Zhang et al., 1999; Daste et al., 2017). Amiloride inhibits macropinocytosis by blocking the $Na^+/H^+$ exchanger, resulting in acidification of the submembranous space and inhibition of requisite small GTPases (Koivusalo et al., 2010). We first tested the impact of pre-treatment with amiloride (1 h; 300 µM) or vehicle (DMSO) on ATP-induced internalization of PANX1-EGFP constitutively expressed in murine neuroblastoma N2a cells using an analysis paradigm established in our previous studies (Boyce and Swayne, 2017; Boyce et al., 2015). Following amiloride pre-treatment, there was no change in intracellular PANX1 or cross-sectional area 30 min post-stimulation with 500 µM ATP (Fig. 1C-E). This suggested that amiloride-sensitive macropinocytic GTPases were required for ATP-induced increase in cell size and PANX1 internalization.

ARF6 regulates endocytic cargo transit between the plasma membrane and a clathrin-independent endosomal compartment that can mature into a macropinosome (Honda et al., 1999; Brown et al., 2001). To further validate that the ATP-evoked increase in cell size was a macropinocytic mechanism, we used cyan fluorescent protein (CFP)-tagged wild-type ARF6 or GTP-hydrolysis resistant (i.e. constitutively active) ARF6 Q67L that would constitutively trigger internalization and thereby create a ceiling for increased cell size. We found that cross-sectional cellular area was increased in cells expressing constitutively active ARF6 Q67L, independent of ATP treatment, supporting our finding that ATP-mediated macropinocytosis increases N2a cell area (Fig. 1F,G). We next assessed the role of ARF6 in ATP-dependent PANX1 internalization by co-expressing ARF6-CFP or ARF6 Q67L-CFP with PANX1-RFP in N2a cells. As expected, ARF6 Q67L expression triggered the formation and retention of large intracellular PANX1-positive/EEA1-positive vesicles independent of ATP stimulation (Fig. 1F-H), while ATP-induced PANX1 internalization was not affected by co-expression of the wild-type ARF6 (Fig. 1I).

## PANX1 internalizes to macropinosomes following ATP stimulation

Given that amiloride and expression of constitutively active ARF6 mutant impacted internalization, we sought to further validate our hypothesis that PANX1 was internalized via macropinocytosis. Canonical endosomes (i.e. caveolae and clathrin-coated pits) are small (<0.2 µm diameter) in relation to macropinosomes (>0.2 µm diameter; Swanson and King, 2019). Using the fluorophore tetramethylrhodamine isothiocyanate (TRITC)-conjugated to a 70 kDa dextran (70 kDa-dextran-TRITC) can reliably label putative macropinosomes (Lin et al., 2020), as it is excluded based on size from vesicles endocytosed through canonical endocytic pathways (i.e. clathrin or caveolin-mediated internalization). Cells were fixed after 30 min incubation and imaged using confocal microscopy, and vehicle was compared to inhibitor pre-treatment in ATP-stimulated cells. Here, there was clear evidence of dextran engulfed by PANX1-positive membranous structures (Fig. 2A,B). The distribution of diameters of these PANX1(+), dextran(+), and PANX1-dextran co-(+) structures were similarly distributed and primarily between 0.8-1.8 µm (Fig. 2C), supporting that these were macropinosomes. Next, we applied several inhibitors or vehicle 1 h prior to initiating PANX1 internalization via infusion of ATP (500 µM, 30 min) along with the macropinosome cargo 70 kDa-dextran-TRITC (100 µg/ml) to the culture medium, then quantified both intracellular PANX1, as well as its co-distribution with dextran-TRITC. Ethylisopropyl amiloride (EIPA), an amiloride derivative, inhibits $Na^+/H^+$ exchangers and TRPP3 channels and is a more potent inhibitor of macropinocytosis than amiloride (Koivusalo et al., 2010; Lin et al., 2018). Relative to vehicle, cells treated with EIPA (25 µM) had less internalized PANX1 30 min after ATP stimulation (Fig. 2D,E). Next, we bathed N2a cells in medium containing LY294002 (25 µM), a selective phosphoinositide-3 kinase (PI3K) inhibitor, which disrupts the closure phase of macropinosomes during internalization (Araki et al., 1996). LY294002 also reduced PANX1 internalization following ATP stimulation (Fig. 2D,E). Macropinocytosis is preceded by local actin-dependent filopodial dynamics prior to internalization and actin also plays an important role in closing the macropinosome following internalization (Swanson, 2008). Thus, to further support our hypothesis that PANX1 is internalized via macropinocytosis, we disrupted actin polymerization using latrunculin A (15 µM, LatA). As with EIPA and LY294002, ATP-induced PANX1 internalization was significantly reduced following LatA pre-treatment (Fig. 2D,E). In support of our findings above, PANX1 distribution to 70 kDa dextran(+) structures was reduced in the presence of all macropinocytosis-targeting inhibitors (EIPA, LY294002, and LatA; Fig. 2F).

## PANX1 C-terminus interacts with lipids involved in macropinocytosis

The carboxy-tail of PANX1 (PANX1$^{CT}$) is the location of PANX1-actin protein-protein interactions (Wicki-Stordeur and Swayne, 2013; Bhalla-Gehi et al., 2010), regulates internalization and cell surface trafficking (Gehi et al., 2011; Bhalla-Gehi et al., 2010), and contains a highly-disordered putative membrane-associated region (Spagnol et al., 2014; Fig. 3A) that was unresolved in cryo-EM structures. Recent atomic resolution structures identified PANX1-lipid interactions (Ruan et al., 2020; Kuzuya et al., 2022), with functional studies suggesting that lipid subtypes might permeate PANX1 (Anderson et al., 2024 preprint; Bialecki et al., 2020) and regulate channel gating (Kuzuya et al., 2022; Henze et al., 2025). Lipids play a key role in many aspects of endocytosis, from enabling cargo recognition to including shaping endocytic membrane structures allowing for internalization, and facilitating appropriate interactions with intracellular signalling molecules that affect internalization and trafficking (reviewed in Sapon et al., 2023; Ewers and Helenius, 2011; Groza et al., 2024; Posor et al., 2022; Wang et al., 2019). As PANX1 internalization was cholesterol dependent (Boyce and Swayne, 2017; Boyce et al., 2015), we

Biology Open

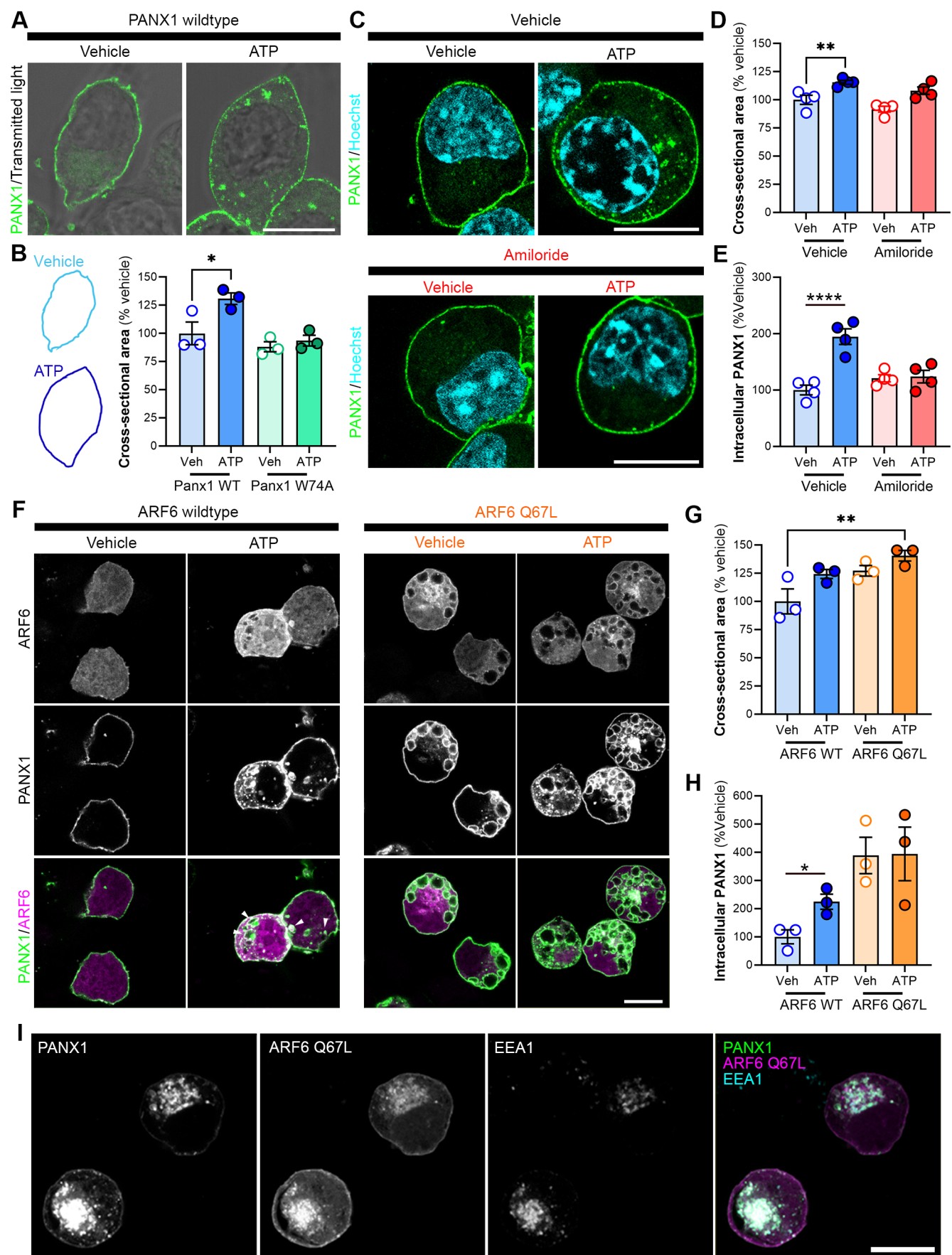

**Fig. 1.** See next page for legend.

**Fig. 1. Elevated extracellular ATP increases N2a cell size through a putative PANX1- and macropinocytosis-dependent mechanism.**
(A) Representative confocal micrographs of PANX1-EGFP (green) N2a cells overlaid with brightfield imaging following incubation with vehicle (water) or ATP (500 µM; 30 min). (B) Quantification of cross-sectional cellular area in PANX1-EGFP (representative outlined traces) or PANX1 W74A-EGFP N2a cells following treatment with vehicle (water) or ATP (500 µM). One-way ANOVA with Dunnett's *post hoc* (N=3). (C) Representative confocal micrographs of PANX1-EGFP (green) N2a cells pre-treated with vehicle (DMSO), or amiloride (300 µM) for 1 h prior to vehicle (water) or ATP (500 µM; 30 min). (D,E) Cross-sectional cellular area (D) and intracellular PANX1 (E) was quantified relative to vehicle control. N=4, two-way ANOVA with Dunnett's *post hoc* [interaction: $F_{(1,12)}=19.53$, $P=0.0008$; inhibitor: $F_{(1,12)}=5.825$, $P=0.0327$; ATP: $F_{(1,12)}=22.22$, $P=0.0005$]. Hoechst (cyan) was used as a nuclear counterstain. (F) Representative confocal micrographs of CHX-treated PANX1-RFP (green) N2a cells transiently transfected with ARF6-CFP or ARF6-Q67L-CFP (magenta, 48 h) prior to vehicle (water) or ATP (500 µM; 30 min), indicating accumulation of intracellular PANX1 with co-expression of constitutively active ARF6 mutant. White arrowheads indicate overlap of internalized vesicles. (G,H) Cross-sectional cellular area (G) and intracellular PANX1 (H) was quantified for each ARF6 construct relative to vehicle treatment. N=3, (unpaired *t*-tests, WT: $P=0.026$, Q67L: $P=0.96$).
(I) Representative confocal micrographs demonstrating accumulation of PANX1RFP (green) and ARF6-Q67L-CFP (magenta) in early endosomes (EEA1, cyan). N refers to the number of coverslips from independent passages (≥30 cells analysed per coverslip). Individual data points are represented with mean±s.e.m. *P<0.05, **P<0.01, ****P<0.0001. Scale bars: 10 µm.

hypothesized that the PANX1[CT] might interact with cholesterol or other lipids involved in cholesterol-dependent and/or clathrin-independent endocytic processes (Sapon et al., 2023; Ewers and Helenius, 2011; Posor et al., 2022; Wang et al., 2019). For example, cholesterol and $PI(4,5)P_2$ are enriched at actin-rich plasma membrane ruffles associated with macropinocytosis (Grimmer et al., 2002). Here, membrane ruffle-localized ARF6 activates PI4P5 kinase (Naslavsky et al., 2003) to generate $PI(4,5)P_2$ (Honda et al., 1999; Brown et al., 2001). Generation of $PI(4,5)P_2$ enables recruitment of the small GTPases Rac1 and Cdc42 to initiate macropinocytic internalization. To test this prediction, we incubated purified GST-tagged PANX1[CT] or GST alone with a hydrophobic membrane spotted with 15 different membrane lipids. After washing off excess unbound PANX1[CT], we probed the membrane using an anti-GST (Fig. 3B) or anti-Panx1[CT] antibody (Fig. 3C; Fig. S1). We did not detect an enriched interaction between PANX1[CT] and cholesterol; however, we found that PANX1[CT] interacted with PA, PI(4)P, and $PI(4,5)P_2$, known regulators of macropinocytosis (Honda et al., 1999; Brown et al., 2001).

## DISCUSSION

Precise control of purinergic signalling via PANX1 is critical in many diverse cell types, including maintaining neural stem cell populations (Wicki-Stordeur and Swayne, 2013; Wicki-Stordeur et al., 2012, 2016), regulating the development of immature neurons (Sanchez-Arias et al., 2020b, 2019), and tumorigenicity and metastasis of several cancers (Penuela et al., 2012; Freeman et al., 2019; Furlow et al., 2015; Di Virgilio et al., 2018). Macropinocytosis is also intimately involved in regulating these processes (Finicle et al., 2018; Bloomfield and Kay, 2016). ATP-mediated macropinocytosis might not only change the receptor contribution but could also be exploited as an adaptive response for uptake of ATP and other nutrients in cells with elevated energy requirements [i.e. cancer (Finicle et al., 2018, Commisso et al., 2013)]. Macropinocytosis activity is elevated at the core of tumours, where nutrients are least concentrated (Lee et al., 2019). In several

types of human lung cancer (as well as breast, liver, and pancreatic cancers), macropinocytosis is used as a mechanism of uptake for ATP, seen by colocalization of fluorescently tagged ATP and 70 kDa dextran (Wang et al., 2017; Recouvreux and Commisso, 2017; Qian et al., 2016; Cao et al., 2019). In that context, internalized ATP is used as an energy source and promotes metastasis in nutrient-starved cancer cells (Cao et al., 2019). Interestingly, PANX1 has also been implicated in the uptake of ATP in both human and yeast cells (Forte et al., 2019); however, it was not determined whether this involved a channel- or internalization-based mechanism. Moreover, constitutive macropinocytosis promotes cell proliferation in many forms of cancer (Stow et al., 2020).

In previous work, we identified that PANX1 internalization occurs in response to elevated extracellular ATP in a dose-dependent manner via interaction with the ionotropic purinergic receptor P2X7R (Boyce and Swayne, 2017; Boyce et al., 2015). This interaction was disrupted by alanine substitution of an extracellular tryptophan (W74A), which in turn inhibited PANX1 internalization. Here, we observe that elevated extracellular ATP also increases the size of mouse neuroblastoma N2a cells. PANX1 W74A, which disrupts the ATP-dependent interaction with P2X7R and downstream internalization (Boyce and Swayne, 2017; Boyce et al., 2015), eliminated the increase in mouse neuroblastoma N2a cell size triggered by extracellular ATP. Cryo-EM-based structures of PANX1 with atomic resolution demonstrated that W74 and its neighbouring residue R75 form the anionic selectivity filter for PANX1 (Ruan et al., 2020; Deng et al., 2020; Mou et al., 2020; Michalski et al., 2020). This filter is likely formed through the formation of a cation-π interaction that stabilizes both large and small anionic molecules as they move through the narrowest portion of the pore (Michalski et al., 2020). We and other have also demonstrated that mutation of this residue disrupts ATP-mediated PANX1 inhibition (Qiu et al., 2012) and P2X7R-interaction dependent internalization (Boyce and Swayne, 2017; Boyce et al., 2015).

ATP-induced increases in cell size, a common consequence of macropinocytosis (Yoshida et al., 2018; Xu et al., 2025), were inhibited by amiloride and facilitated by a constitutively active mutant of the GTPase ARF6. ARF6 is a critical macropinocytosis effector and a driver of macropinosome trafficking following internalization (Honda et al., 1999; Brown et al., 2001). Elevated ATP did not further increase intracellular PANX1 or cell size in the presence of ARF6 Q67L, suggesting that membrane internalization was saturated under these conditions. Here, each manipulation to disrupt ATP-dependent increases in cell size also disrupted PANX1 internalization, suggesting that ATP-induced PANX1 internalization occurs through macropinocytosis. In support of this, PANX1 internalization was sensitive to inhibition of amiloride-sensitive GTPases, PI3K, and actin, and internalized PANX1 had strong co-distribution with the macropinosome cargo 70 kDa dextran following ATP stimulation. Notably, novel PANX1[CT] lipid interactors, $PI(4,5)P_2$ and PA, are produced in response to ARF6 activation and are critical for multiple stages of macropinocytosis (Honda et al., 1999; Brown et al., 2001). Given the role of ARF6 as a macropinsome effector and the sensitive of cell size perturbations to amiloride, these results support that ATP-evoked increases in cell size occur in a PANX1- and macropinocytosis-dependent mechanism. It is not yet clear whether PANX1 W74A disruption of ATP-induced cell area expansion implicates PANX1 directly (either through channel function, protein-protein interactions, or another mechanism) or indirectly

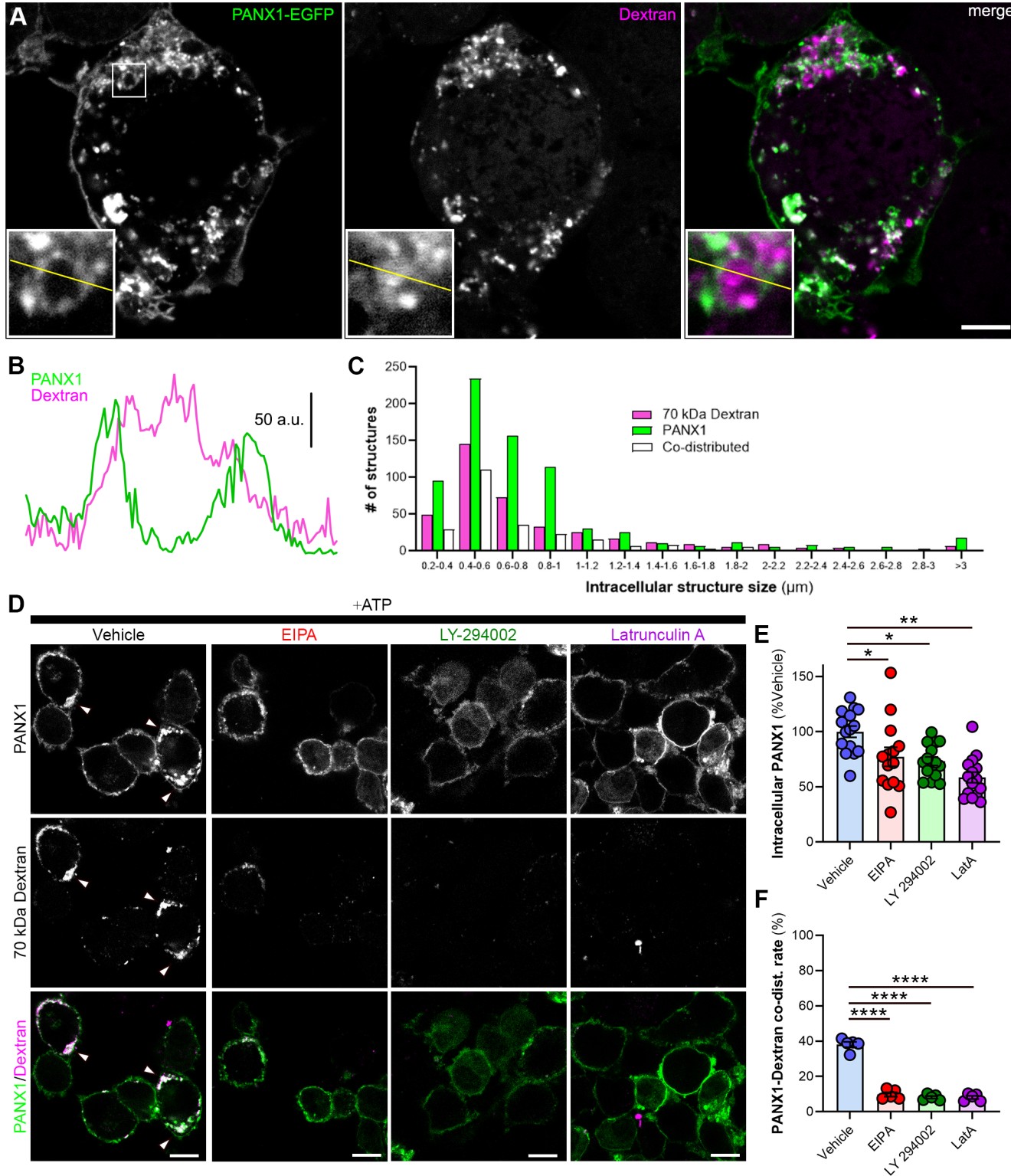

**Fig. 2. Intracellular PANX1 distributes to macropinosomes.** (A) Representative confocal micrographs with line scan over ROI demonstrate PANX1 localization in membrane-like structures surrounding macropinosome cargo 70 kDa dextran-TRITC. Inset highlights putative PANX1-positive macropinosome. (B) Line scan (10 μm) of putative macropinosome with PANX1-encompassing dextran-TRITC cargo. (C) Diameter of intracellular PANX1, TRITC-Dextran, and co-distributed structures show similar size distribution pattern. (D) Representative confocal micrographs of transiently expressed PANX1-EGFP (green) N2a cells pre-treated with vehicle or EIPA (25 μM), LY-294002 (25 μM), or latrunculin-A (15 μM) then incubated with 70 kDa-dextran-TRITC (100 μg/ml, magenta) and ATP (500 μM, 30 min) prior to fixation. (E) Intracellular PANX1 following ATP stimulation (30 min) was quantified relative to vehicle control. $N$=14-15 per treatment group, one-way ANOVA with Dunnett's *post hoc* [interaction: $F(3,55)$=9.124, $P$<0.0001]. (F) Co-distribution rate of intracellular PANX1 with TRITC-dextran following ATP stimulation for each treatment ($N$=5 per treatment group, one-way ANOVA with Dunnett's *post hoc* [interaction: $F(3,16)$=164, $P$<0.0001]. *$P$<0.05, **$P$ 0.01, ****$P$<0.0001. $N$ refers to the number of coverslips from independent passages (≥30 cells analyzed per coverslip). Individual data points are represented with mean±s.e.m. Scale bars: 10 μm.

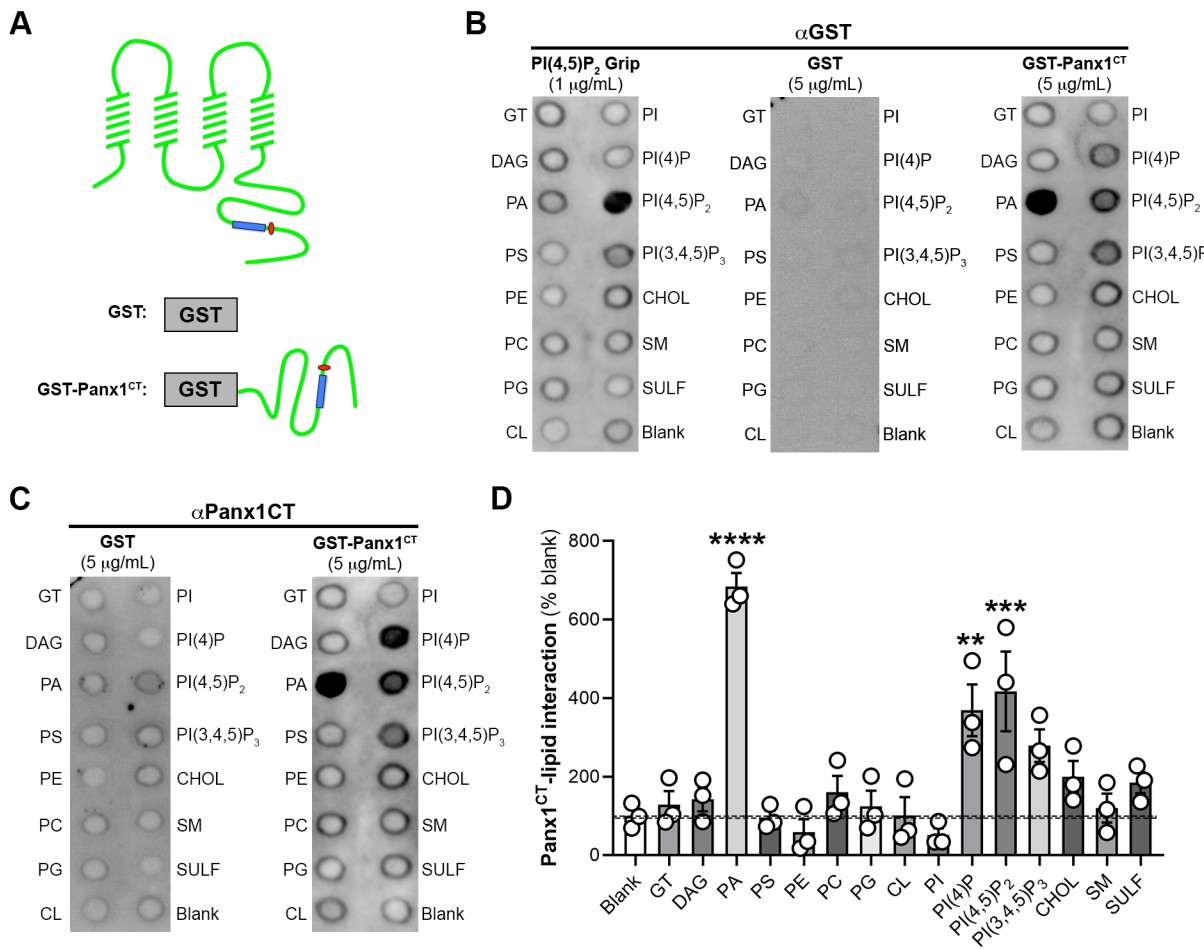

**Fig. 3. The PANX1 C-terminus interacts with phospholipids involved in macropinocytosis.** (A) Schematic of full-length PANX1, as well as GST and GST-fused PANX1 C-terminus (GST-PANX1$^{CT}$), indicating location of putative lipid interaction domain (blue rectangle) relative to the caspase cleavage site (red circle). (B) Western blot of Membrane Lipid Interaction Strip (hydrophobic membrane spotted with 15 different membrane lipids [GT – glyceryl tripalmitate, DAG – diacylglycerol, PA – phosphatidic acid, PS – phosphatidylserine, PE – phosphatidylethanolamine, PC – phosphatidylcholine, PG – phosphatidylglycerol, CL – cardiolipin, PI – phosphatidylinositol, PI(3)P – phosphatidylinositol 3-phosphate, PI(3,4)P2 – phosphatidylinositol 3,4-bisphosphate, PI(3,4,5)P3 – phosphatidylinositol 3,4,5-triphosphate, CHOL – cholesterol, SM – sphingomyelin, SULF – 3-sulfogalactosylceramide] and a blank (xylene cyanol FF) incubated with purified peptides including positive control [PI(4,5)P2 Grip (1 μg/ml)], GST (5 μg/ml) or GST-PANX1$^{CT}$ (5 μg/ml) and probed with GST antibody. (C) Western blot of Membrane Lipid Interaction Strip that was pre-incubated with purified GST (5 μg/ml) or GST-PANX1$^{CT}$ (5 μg/ml) and probed with PANX1CT antibody. (D) Relative GST-PANX1$^{CT}$ lipid interaction enrichment on Western blot, normalized to blank, was quantified using densitometry. $N=3$, one-way ANOVA with Dunnett's *post-hoc* (**$P<0.01$, ***$P<0.001$, ****$P<0.0001$). $N$ refers to the number of coverslips from independent passages ($\geq$ 30 cells analyzed per coverslip). Individual data points are represented with mean±s.e.m.

(such as via the loss of an intracellular function) in the regulation of ATP-mediated macropinocytosis, but both may be possible. Manipulation of PANX1 expression could be an important next step for understanding its role in macropinocytosis.

In addition to regulating purinergic signalling pathways, it is tempting to speculate that ATP release by PANX1 may, in part, act in a homeostatic autocrine manner to normalize cell size. PANX1 demonstrates cell type-specific mechanosensitivity (Nielsen et al., 2020; Bao et al., 2004; Seminario-Vidal et al., 2011; Locovei et al., 2006), where changes in cell volume or mechanical stress activate PANX1 channels. In oocytes and HEK293T cells, osmotic shrinkage activates PANX1-mediated membrane currents (but no dye uptake), yet swelling does not impact PANX1 function (Nielsen et al., 2020). While in airway epithelia, hypotonic challenge stimulated ATP release and dye uptake (Seminario-Vidal et al., 2011). Not only could PANX1 impact overall cell size in this manner, but there is also potential that, once internalized, PANX1 could regulate endosomal volume. Channel-dependent Cl$^-$ efflux from endosome compartments was

recently demonstrated to regulate endosome volume (Freeman et al., 2020). The anion-selective PANX1 may be able to regulate endosome volume, if active, following internalization. Future investigations of the functional consequences of the relationship between PANX1 mechanosensation and macropinocytosis, and the function of endosome-resident PANX1 could resolve these outstanding questions.

There are several outstanding questions with regards to the regulation of PANX1 trafficking, particularly with regards to lability of cell surface PANX1 to internal and external stimuli and the destination and function of internalized PANX1. The anterograde trafficking mechanisms of PANX1 have been clearly delineated (Boyce et al., 2014), where PANX1 receives a series of post-translational N-linked glycosylation modifications as it passes through the endoplasmic reticulum and Golgi on its way to the cell surface (Penuela et al., 2007, 2008, 2009). Once at the cell surface, an interaction between PANX1 and actin regulates stability (Bhalla-Gehi et al., 2010). Here, the balance between PANX1 endocytosis, trafficking, recycling, and degradation is worthy of further exploration.

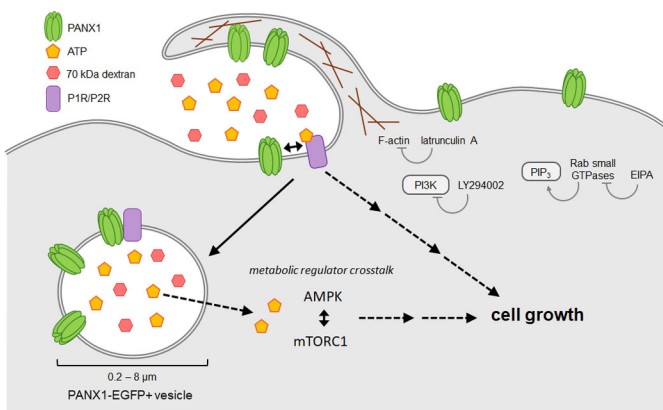

**Fig. 4. ATP increases murine neuroblastoma cell size through a PANX1- and macropinocytosis-dependent mechanism.** ATP evokes internalization of PANX1 via macropinocytosis with a PANX1-dependent increase in cell size. PANX1 internalization was disrupted through inhibition of actin polymerization, PI3K, and pH sensitive GTPases. The increase in cell size due to elevated extracellular ATP could arise from multiple, possibly convergent pathways, such as purinergic receptor signalling, and/or crosstalk between metabolic master regulators mTORC1 and AMPK.

PANX1 has a half-life of >24 h (Bhalla-Gehi et al., 2010; Boassa et al., 2008), and there is evidence for endolysosomal degradation under naïve conditions (Gehi et al., 2011). In response to ATP, internalized PANX1 was enriched in Rab14-positive recycling endosomes (Boyce et al., 2015). To date, PANX1 internalization in several cell types has pointed to a predominantly clathrin-independent mechanism (Boyce and Swayne, 2017; Boyce et al., 2015; Gehi et al., 2011); however, this does not rule out the possibility that PANX1 is internalized through a combination of clathrin-dependent and independent mechanisms depending on contextual cues (i.e. varying stimuli, lipid dynamics, tissue type), as has been observed for other plasma membrane proteins (Sandvig et al., 2018; Lundmark and Carlsson, 2010; Grant and Donaldson, 2009). Further, internalized plasma membrane cargo undergoes an ever-shifting balance between degradation and recycling. Thus, future work could further explore if PANX1 trafficking mechanisms are stable across cell types or specific to different tissues or contexts.

We identified several lipid interactions at the PANX1 C-terminus (Fig. 3), each playing an important role in endocytic pathways, including macropinocytosis (Honda et al., 1999; Brown et al., 2001). The generation of PI(4,5)P$_2$-rich membrane domains, from its precursor PI(4)P, create hotspots for macropinocytosis initiation. Both of these lipids had significant interactions with the PANX1$^{CT}$ and will be the subject of future investigations (Grimmer et al., 2002). Lipid subtypes may also regulate PANX1 channel gating (Kuzuya et al., 2022; Henze et al., 2025) and even permeate the channel (Anderson et al., 2024 preprint; Bialecki et al., 2020). Whether the lipid interactors identified here are involved in these processes is an open question.

Taken together, our data suggest that elevated extracellular ATP drives a macropinocytosis- and PANX1-mediated increase in cell size (Fig. 4). PANX1 internalization and cell size increases normally induced by elevated extracellular ATP are absent with PANX1 W74A, suggesting that PANX1 may also directly or indirectly regulate ATP-evoked macropinocytosis and, consequently, cell size. The coordinated relationship between PANX1 and macropinocytosis has implications for many cellular behaviours where purinergic signalling is involved, particularly those, like cancer, where ATP can act as both a signalling molecule in the extracellular space and as a

metabolite supporting enhanced cell growth when internalized. This highlights another possible mechanism for PANX1 to promote a survival advantage for cancer cells.

## MATERIALS AND METHODS

### Plasmids
The PANX1-EGFP and PANX1-RFP plasmids (Penuela et al., 2007) were generous gifts from Drs Dale Laird and Silvia Penuela. The pARF6-CFP (Plasmid #11382) and pARF6 Q67L-CFP (Plasmid #11387) were acquired from Addgene, courtesy of Dr Joel Swanson.

### Cell culture
Neuro2a (N2a) mouse neuroblastoma cells (procured from the American Type Culture Collection) were cultured in Dulbecco's modified Eagle's medium (DMEM)/F12 supplemented with 10% FBS, 100 units/ml penicillin, and 100 mg/ml streptomycin (all obtained from Gibco/Life Technologies) and routinely assessed for contamination. Where indicated, N2a cells were transfected using jetPEI reagent (Polyplus transfection/VWR) according to the manufacturer's protocol. N2a cells stably or transiently expressing PANX1-EGFP or PANX1-RFP were maintained in DMEM/F12 containing 10% FBS, and 100 units/ml penicillin, 100 μg/ml streptomycin, and 400 μg/ml geneticin 418 (all obtained from Gibco/Life Technologies). Stable cell lines were generated as follows: N2a cells were transfected with PANX1-EGFP or PANX1-RFP using the jetPEI reagent (Polyplus transfection/VWR) according to the manufacturer's protocol. Cells were plated on poly-D-lysine (Sigma-Aldrich)-coated coverslips. For all internalization-related experiments, protein translation was briefly inhibited by treatment with 20 μg/ml cycloheximide (CHX; Sigma-Aldrich) for 8 h, coincident with other treatments. Where indicated, stable cell lines were transfected with ARF6-ECFP or ARF6-Q67L-ECFP, Cells were fixed with 4% paraformaldehyde (PFA) and washed three times in phosphate buffered saline (PBS), prior to mounting for imaging or processing for immunostaining. During experimentation, the investigator was unaware of group allocation. We investigated the role of ATP (500 μM; Sigma-Aldrich) versus vehicle control (equal volume of water) on PANX1 cell surface stability and trafficking. We disrupted macropinocytosis by pre-treating cells with amiloride (300 μM; Sigma-Aldrich), EIPA (25 μM; Tocris) Latrunculin A (15 μM; Sigma-Aldrich), LY 294002 (25 μM; Tocris), or vehicle (DMSO) for 1 h. To identify co-distribution with macropinosome cargo, where indicated, cells were treated with ATP (500 μM; Sigma-Aldrich) and 70 kDa TRITC-Dextran (100 μg/ml; Thermo Fisher Scientific) for 30 min prior to fixation.

### Immunocytochemistry
Antibody labelling of PFA-fixed cultures was performed as previously described (Boyce et al., 2015). The primary antibody used was early endosome antigen 1(EEA1; 1:200; Cell Signaling Technology) and the secondary antibody was Alexa647 AffiniPure donkey anti-rabbit IgG (1:600; Jackson ImmunoResearch).

### Microscopy
Confocal microscopy and analysis were performed unaware of the experimental conditions. Images were acquired with a Leica TCS SP8 confocal STED microscope. Quantification was performed using Leica Application Suite (version 3.1.3) and in the z-section displaying the largest plane of the nucleus (Hoechst 33342), where applicable. In the absence of Hoechst staining (where CFP-tagged constructs were expressed), a z-stack was captured over the entirety of the cells in the field of view and the middle z-plane was selected for region of interest (ROI) analysis. ROIs with a cross-sectional area of ≤60 mm (Steinman et al., 1976) were excluded from analysis (three ROIs total, two from ARF6-CFP – vehicle, one from ARF6 Q67L-CFP – vehicle). Comparisons were made between images acquired under identical conditions. Representative confocal micrographs were adjusted for contrast uniformly using Adobe Photoshop (CC 2015.1.2) for display purposes only; no contrast adjustments were made prior to analysis. Confocal images (Leica TCS SP8) of fixed cells were acquired using a 40× (1.3 NA) oil immersion objective at 3× optical zoom in 1296×1296 format

Biology Open

with a pixel area of 71 nm (Steinman et al., 1976) as confocal z-stacks. Quantification of PANX1-EGFP/PANX1-RFP fluorescence intensity to describe 'intracellular PANX1' was performed at time zero and at 30 min post stimulation, as follows: a polygonal trace was drawn 1 μm inside the cell periphery and the encapsulated average PANX1 fluorescence intensity per pixel was computed; quantification of cross-sectional cellular area was determined on the same z-plane as the intracellular area; cross-sectional area per cell was the area encapsulated by the ROI traced along the cell periphery, as described above.

## Intracellular structure analysis

To measure the size of potential macropinosomes, PANX1-EGFP signals were acquired with confocal microscopy using a 100× (1.3 NA) oil immersion objective in a 6008×6008 format, yielding a pixel size of 19 nm, and the pinhole size was set to 0.65 arbitrary units (a.u.). Inclusion criteria for macropinosomes were round structures with a diameter of 0.2–5 μm that were positive for both PANX1-EGFP and/or TRITC-dextran. The diameters of the selected macropinosomes were measured using ImageJ (Version 2.14.0). The investigator was unaware of experimental conditions during analysis.

## Co-distribution analysis

For co-distribution analysis of PANX1 and 70 kDa dextran, confocal images of fixed PANX1-EGFP N2a cells treated with 70 kDa dextran-TRITC and ATP (500 μM, 30 min) were obtained using the Leica TCS SP8 with a 40× (1.3 NA) oil immersion objective at 2× optical zoom in a 1904×1904 format, resulting in a pixel area of 76.22 μm² as confocal z-stacks. A z-stack of three planes (z-step size set to 0.1 μm) was captured over the plane with the largest nucleus in the field of view. The z-plane displaying the largest plane of the nucleus among the three planes was selected for ROI analysis. The selected plane contained at least 15 PANX1-EGFP[+] cells. To quantify colocalization using Leica Application Suite, the threshold for both the intracellular PANX1 and dextran intensity was set to 30%. Colocalization rate by computing the fluorescence intensity per pixel. The investigator was unaware of experimental conditions during analysis.

## Membrane lipid strip interaction assays

Membrane Lipid Strips (Echelon Biosciences; hydrophobic membrane spotted with 15 different membrane lipids [GT – glyceryl tripalmitate, DAG – diacylglycerol, PA – phosphatidic acid, PS – phosphatidylserine, PE – phosphatidylethanolamine, PC – phosphatidylcholine, PG – phosphatidylglycerol, CL – cardiolipin, PI – phosphatidylinositol, PI(4)P – phosphatidylinositol 4-phosphate, PI(4,5)P$_2$ – phosphatidylinositol 4,5-bisphosphate, PI(3,4,5)P$_3$ – phosphatidylinositol 3,4,5-triphosphate, CHOL – cholesterol, SM – sphingomyelin, SULF – 3-sulfogalactosylceramide] and a blank (xylene cyanol FF) were blocked for 1 h in blocking buffer (3% BSA in PBS-T). Strips were then transferred to blocking buffer containing purified protein of interest: PI(4,5)P$_2$ Grip (1 μg/ml, positive control from Echelon Biosciences), GST (5 μg/ml), or GST-fused PANX1 C-terminus (GST-PANX1$^{CT}$, 5 μg/ml) for 1 h, then washed three times in PBS-T and incubated for an additional hour in blocking buffer with primary antibody. Strips were again washed in PBS-T three times then incubated in corresponding HRP-conjugated secondary for 1 h. All incubations were performed at room temperature with gentle agitation. Lipid strips were probed using the primary GST antibody, then stripped and re-probed using the primary PANX1 antibody (see Fig. S2 for details).

## Statistical analysis

The datasets used and/or analysed during the current study are available from the corresponding author on reasonable request. A biological replicate was defined as a coverslip obtained from an independent cell passage, where a technical replicate is each single cell on a given coverslip. Results were analysed using two-way ANOVA, one-way ANOVA or unpaired t-tests, where applicable. Data are presented as mean±s.e.m. Detailed information about statistical tests is available in the figure legends.

## Acknowledgements

The authors are grateful to Juan C. Sanchez-Arias and Emma van der Slagt for their initial input on study design and analysis not included in the updated version of the manuscript. This manuscript features data published in theses by A.K.J.B. (Boyce, 2017) and H.Y. (You, 2024), and was posted as a bioRxiv preprint (doi:10.1101/2025.09.11.675618).

## Competing interests

The authors declare no competing or financial interests.

## Author contributions

Conceptualization: A.K.J.B., L.A.S.; Data curation: A.K.J.B., H.Y.; Formal analysis: A.K.J.B., L.E.W.-S.; Funding acquisition: L.A.S.; Investigation: A.K.J.B., H.Y., L.E.W.-S.; Methodology: A.K.J.B.; Project administration: L.A.S.; Resources: L.A.S.; Supervision: L.A.S.; Visualization: A.K.J.B., L.E.W.-S.; Writing – original draft: A.K.J.B., L.A.S.; Writing – review & editing: A.K.J.B., L.E.W.-S., L.A.S.

## Funding

This work was supported by operating grants from the Natural Sciences and Engineering Research Council of Canada (NSERC; RGPIN-2017-03889), the Canadian Institutes of Health Research (MOP142215; PJT 189953), and the University of Victoria-Division of Medical Sciences to L.A.S.. L.A.S. was also supported by a Michael Smith Foundation for Health Research and British Columbia Schizophrenia Society Foundation Scholar Award (5900). A.K.J.B. was supported by scholarships from NSERC (PGSD 459931-2014) and the University of Victoria (President's Research Scholarship, Dr Howard E. Petch and Dr Julius F. Schleicher Memorial Scholarships) for this work and is currently supported, in part, by the National Institute of General Medical Sciences of the National Institute of Health under award P20GM109089. H.Y. was supported by an NSERC Canada Graduate Research Scholarship – Master's program. L.A.S. is also grateful for infrastructure support from the Canada Foundation for Innovation (29462) and the British Columbia Knowledge Development Fund (804754) for the Leica SP8 microscope system. Open Access funding provided by University of New Mexico. Deposited in PMC for immediate release.

## Data and resource availability

All relevant data and details of resources can be found within the article and its supplementary information.

## Peer review history

The peer review history is available online at https://journals.biologists.com/bio/lookup/doi/10.1242/bio.062272.reviewer-comments.pdf

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
