## [Peer Review File · Biology Open]

ATP increases murine neuroblastoma cell size through a PANX1- and macropinocytosis-dependent mechanism

Haifei You, Leigh E. Wicki-Stordeur, Leigh Anne Swayne and Andrew KJ Boyce
DOI: 10.1242/bio.062272

Editor: Catherine L. Jackson

Review timeline

Submission to sister journal:	13 August 2025
Editorial decision at sister journal:	21 August 2025
Transfer to Biology Open:	24 September 2025
Editorial decision:	20 October 2025
First revision received:	30 October 2025
Accepted:	31 October 2025

Original submission to sister journal

First decision letter

MS Title: ATP increases murine neuroblastoma cell size through a PANX1- and macropinocytosis-dependent mechanism

Authors: Andrew KJ Boyce; Haifei You; Leigh E. Wicki-Stordeur; Leigh Anne Swayne
Article Type: Research Article

Thank you for your recent submission. Unfortunately, we feel the manuscript would be inappropriate for this journal. Our instructions to referees indicate that we will only accept articles that provide mechanistic insight into fundamental cellular processes. Although the work is interesting and appears to have been carried out carefully, it does not satisfy these criteria. The comments of one independent expert referee who was consulted about the manuscript are provided below.

Reviewer 1

Comments for the author

This manuscript addresses the route of internalization of pannexin 1. The authors describe the potentially interesting finding that pannexin1 is internalized by macropinocytosis stimulated by ATP. They also describe that the C-terminus of pannexin 1 interacts with a number of lipids and link pannexin 1 and macropinocytosis to an increase in cell size. Unfortunately even for a Short Report the results are quite preliminary and more extensive characterization would be required to confirm the reported findings of the study.

All experiments use overexpressed Panx1 which shows high uniform PM labeling. This raises the question of whether ATP-stimulated macropinocytosis actually reflects specific endocytosis of Panx1 or just reflects inclusion into the major pathway induced by ATP, macropinocytosis; would other surface proteins expressed at similar levels show inclusion into macropinosomes in response to stimulation? The authors need to look at endogenous Panx1 protein and/or the tagged protein at endogenous levels and to test the specificity of the induced macropinocytosis on Panx1.

The authors should test knockdown of key macropinocytosis regulators such as CTBP1, Rac1, and Rabankyrin-5, not just rely on overexpression of an ARF6 mutant. Macropinocytosis should also be quantitated.

The effects of a point mutation in Panx1 are interesting but the authors should also test the effect of loss or knockdown of Panx1 on macropinocytosis.

The observation of binding to specific lipids is of potential interest but is not connected to the rest of the paper and the suggestion that these findings are linked to macropinocytosis is not tested.

The links to cell size are not well characterised.

Transfer to Biology Open

Author response to reviewers' comments

Response: Thank you for your encouragement to submit this article to Biology Open, we have addressed several concerns, included additional analyses and rewritten sections of the paper prior to transfer.

Reviewer 1

This manuscript addresses the route of internalization of pannexin 1. The authors describe the potentially interesting finding that pannexin 1 is internalized by macropinocytosis stimulated by ATP. They also describe that the C-terminus of pannexin 1 interacts with a number of lipids and link pannexin 1 and macropinocytosis to an increase in cell size. Unfortunately even for a Short Report the results are quite preliminary and more extensive characterization would be required to confirm the reported findings of the study.

Response: We thank the reviewer for their important criticisms, which have been addressed to the best of our ability in the updated manuscript (highlighted in yellow) submitted to Biology Open.

Comments

All experiments use overexpressed Panx1 which shows high uniform PM labeling. This raises the question of whether ATP-stimulated macropinocytosis actually reflects specific endocytosis of Panx1 or just reflects inclusion into the major pathway induced by ATP, macropinocytosis; would other surface proteins expressed at similar levels show inclusion into macropinosomes in response to stimulation? The authors need to look at endogenous Panx1 protein and/or the tagged protein at endogenous levels and to test the specificity of the induced macropinocytosis on Panx1.

Response: This is an important consideration that has been addressed in our revised submission, as well as in previous manuscripts using the same stable PANX1-EGFP expressing N2a cell line for internalization experiments. In our current manuscript, we have included additional references highlighting that the expression of PANX1-EGFP is similar to endogenous PANX1 (lines 238-242). In creating this stable cell line, we selected for low PANX1-EGFP expression to approach the expression level of endogenous PANX1 (see description in Boyce et al., 2015 also line 238 in our updated manuscript). In Boyce et al. Biochem J, 2015, we performed live imaging with wheat germ agglutinin labeled membrane. Importantly, we demonstrated that there were PANX1-EGFP both PANX1-positive and PANX1-negative endosomal compartments suggesting that PANX1 internalization by ATP was a selective process, rather than PANX1 being included in a major pathway induced by ATP. Further, we believe that the inhibition of ATP-mediated PANX1 internalization by alanine-substitution of the extracellular residue PANX1 W74A further supports specificity (Boyce et al., 2015 & 2017, Figure 1).

The authors should test knockdown of key macropinocytosis regulators such as CTBP1, Rac1, and Rabankyrin-5, not just rely on overexpression of an ARF6 mutant. Macropinocytosis should also be quantitated.

Response: In lieu of knockdown experiments, we use pharmacology to target micropinocytosis related proteins (see amiloride Figure 1C-D, see EIPA, LY-294002, and Latrunculin A Figure 2). To quantify micropinocytosis, in Figure 2, the diameter of internalized structures containing the macropinosome cargo 70 kDa Dextran-TRITC and PANX1-EGFP are quantified, highlighting that most exist as the expected size of macropinosomes. Next, in ATP-treated N2a cells, we demonstrate the PANX1 codistribution with 70 kDa Dextran-TRITC is reduced in cells treated with EIPA, LY-294002, and LatA (relative to vehicle). We believe this is sufficient to identify macropinocytosis as the mechanism for PANX1 internalization.

The effects of a point mutation in Panx1 are interesting but the authors should also test the effect of loss or knockdown of Panx1 on macropinocytosis.

Response: While this would provide support for our findings, we believe it is outside the scope of the current manuscript and we plan to tackle this in a separate, but related project. PANX1 expression is developmentally regulated and we are currently exploring how this might impact macropinocytosis. We have also added a sentence to the discussion highlighting this as an important next step (lines 376-377).

The observation of binding to specific lipids is of potential interest but is not connected to the rest of the paper and the suggestion that these findings are linked to macropinocytosis is not tested.

Response: While outside the scope of the current manuscript, we have added sections to the introduction and the discussion to better incorporate our interpretation of this exciting lipid interaction data and what it could mean for next steps in our labs and in the field (lines 126-130; 390-396).

The links to cell size are not well characterised.

Response: The manuscript has been reorganized to better emphasize the impact of extracellular ATP on cell size. Here, cell size comparisons have been moved from their own figure at the end of the paper (Figure 4) and integrated into the paper from the start (see Figure 1). The manuscript text has been revised to reflect this change. Further, new data has been added describing the methodology for cell size quantification (Figure 1A & B). Here, we believe that extracellular ATP is triggering an increase in neuroblastoma cell size through a mechanism that relies on PANX1-mediated macropinocytosis.

Overall, the findings are interesting but too preliminary for JCS, even as a short report.

Response: We believe that the updated manuscript introduces ATP-evoked increases in neuroblastoma cell size, provides a clear mechanistic understanding of PANX1 internalization and its link to cancer cell behavior, and exciting lipid interaction data which will be the target of future experimentation.

First decision letter

MS ID#: bio.062272

MS Title: ATP increases murine neuroblastoma cell size through a PANX1- and macropinocytosis-dependent mechanism

Authors: Andrew KJ Boyce; Haifei You; Leigh E. Wicki-Stordeur; Leigh Anne Swayne

Article Type: Research Article

I have now reached a decision on the above manuscript.

The reviewer reports are shown at the bottom of this email or can be accessed, together with a copy of this decision letter, by going to:

As you will see, the reviewers gave favourable reports, but raised a few small points that will require amendments to your manuscript. I hope that you will be able to carry these out, because we would like to be able to accept your paper.

At this stage, we also ask you to ensure your manuscript complies with our formatting guidelines - please see our manuscript preparation guidelines for details. Provided you are able to fully address the referees' comments, we are positive about publication of your paper (we accept over 95% of revision submissions) and therefore hope you won't mind any extra work involved in reformatting your manuscript at this point.

Please upload both a 'clean' version of your Word file, along with a highlighted version clearly showing where you have made changes in the revised manuscript. Please avoid using 'Track changes' in Word files as these are lost in PDF conversion.

I should be grateful if you would also provide a point-by-point response detailing how you have dealt with the points raised by the reviewers in the 'Response to Reviewers' box. Please attend to all of the reviewers' comments. If you do not agree with any of their criticisms or suggestions please explain clearly why this is so.

Reviewer 1

Comments for the author

The manuscript by Boyce and colleagues provide novel insights on the route of ATP-driven internalization of PANX1. Authors describe that wild type-PANX1 but not W74A-PANX1 is internalized through macropinocytosis upon ATP stimulation/treatment, and this ATP-driven internalization of PANX1 is inhibited by amiloride pre-treatment. They also utilize pharmacological strategies to target macropinocytosis - EIPA, LY294002 and LatA, and show that pre-treatment of the cells with EIPA, LY294002 or LatA reduces the co-distribution of PANX1 with putative macropinosomes (labelled with 70kDa Dextran). Lastly, they demonstrate that C-terminus of PANX1 interacts with various lipids, most abundantly with PA, PI(4)P, PI(4,5)P2 and PI(3,4,5)P3. Overall, this a well-written manuscript with interesting findings.

Authors improved their manuscript prior to submission to the Biology Open, including their responses to the previous reviewer comments. Minor comment: typo in the name of the PI3K inhibitor in Figure 4 - LY292004. It should be LY294002. Authors should check this and amend accordingly.

Reviewer 2

Comments for the author

This work provides evidence that the PANX1 channel undergoes macropinocytosis from the use of various pharmacological tools. There is evidence that the PANX1 cytoplasmic domain binds to different membrane lipids which may play a role in trafficking but this is ill-defined at present. I have comments that I hope the authors can address:

- 1) I would like to see some estimate of numbers of PANX1-GFP per cell, and the comparison to physiological levels of PANX1. There is currently also lacking blots for PANX1-GFP and PANX1 in the same cell type which would have been nice to see and compare. The numbers of cell surface 'cargo' molecules is important in considering how many PANX1 molecules are rapidly endocytosed in response to ATP stimulation (of P2X7).
- 2) In this field, the variety of membrane trafficking portals that govern plasma membrane cargo entry into the endosome-lysosome network is generally classified into clathrin-dependent endocytosis (CDE) and clathrin-independent endocytosis (CIE) e.g. caveolae, macropinocytosis, lipid rafts. For any single PM cargo, it can be envisioned that there is a balance between CDE and CIE at steady-state, with changes depending on environmental conditions and different stimuli. The

PANX1 model is no different to say EGFR, transferrin and LDL receptors in this context which all display dynamic associations with different endocytic routes which can be modulated by changing signals and lipid dynamics. The authors need to carefully reframe their language and ideas in this context to make it compatible with these ideas. Please see current references such as:

1: Hemalatha A, Mayor S. Recent advances in clathrin-independent endocytosis. *F1000Res*. 2019 Jan 31;8:F1000 Faculty Rev-138.

2: Sandvig K, Kavaliauskiene S, Skotland T. Clathrin-independent endocytosis: an increasing degree of complexity. *Histochem Cell Biol*. 2018 Aug;150(2):107-118.

3: Cárdenas AM, Marengo FD. Rapid endocytosis and vesicle recycling in neuroendocrine cells. *Cell Mol Neurobiol*. 2010 Nov;30(8):1365-70.

4: Lundmark R, Carlsson SR. Driving membrane curvature in clathrin-dependent and clathrin-independent endocytosis. *Semin Cell Dev Biol*. 2010 Jun;21(4):363-70.

5: Grant BD, Donaldson JG. Pathways and mechanisms of endocytic recycling. *Nat Rev Mol Cell Biol*. 2009 Sep;10(9):597-608.

6: Mayor S, Pagano RE. Pathways of clathrin-independent endocytosis. *Nat Rev Mol Cell Biol*. 2007 Aug;8(8):603-12.

3) It is unclear to me how lipid recognition is linked to endocytosis - are the authors addressing a model specifically here? are such lipids enriched in the macropinosomal structure or are these lipids localised randomly in the PM? There needs to be some better thinking here as to how lipid turnover is linked to PANX1 recruitment to macropinosomes.

4) What is the balance between PANX1 endocytosis, trafficking, recycling or degradation? All PM cargo undergoes recycling and degradation depending on the nature of the protein, this is unclear here.

5) Other examples of ligand-stimulated macropinosytosis should be referenced:

1: Basagiannis D, Zografou S, Murphy C, Fotsis T, Morbidelli L, Ziche M, Bleck C, Mercer J, Christoforidis S. VEGF induces signalling and angiogenesis by directing VEGFR2 internalisation through macropinosytosis. *J Cell Sci*. 2016 Nov 1;129(21):4091-4104.

2: Bannach C, Brinkert P, Kühling L, Greune L, Schmidt MA, Schelhaas M. Epidermal Growth Factor Receptor and Abl2 Kinase Regulate Distinct Steps of Human Papillomavirus 16 Endocytosis. *J Virol*. 2020 May 18;94(11):e02143-19.

3: Weerasekara VK, Patra KC, Bardeesy N. EGFR Pathway Links Amino Acid Levels and Induction of Macropinosytosis. *Dev Cell*. 2019 Aug 5;50(3):261-263.

4: Lee SW, Zhang Y, Jung M, Cruz N, Alas B, Comisso C. EGFR-Pak Signaling Selectively Regulates Glutamine Deprivation-Induced Macropinosytosis. *Dev Cell*. 2019 Aug 5;50(3):381-392.e5.

5: Wang HB, Zhang H, Zhang JP, Li Y, Zhao B, Feng GK, Du Y, Xiong D, Zhong Q, Liu WL, Du H, Li MZ, Huang WL, Tsao SW, Hutt-Fletcher L, Zeng YX, Kieff E, Zeng MS. Neuropilin 1 is an entry factor that promotes EBV infection of nasopharyngeal epithelial cells. *Nat Commun*. 2015 Feb 11;6:6240.

6: Balaji K, Mooser C, Janson CM, Bliss JM, Hojjat H, Colicelli J. RIN1 orchestrates the activation of RAB5 GTPases and ABL tyrosine kinases to determine the fate of EGFR. *J Cell Sci*. 2012 Dec 1;125(Pt 23):5887-96.

7: Sánchez EG, Quintas A, Pérez-Núñez D, Nogal M, Barroso S, Carrascosa ÁL, Revilla Y. African swine fever virus uses macropinosytosis to enter host cells. *PLoS Pathog*. 2012;8(6):e1002754.

8: Solis GP, Schrock Y, Hülsbusch N, Wiechers M, Plattner H, Stuermer CA. Reggies/flotillins regulate E-cadherin-mediated cell contact formation by affecting EGFR trafficking. *Mol Biol Cell*. 2012 May;23(10):1812-25.

9: Schmees C, Villaseñor R, Zheng W, Ma H, Zerial M, Heldin CH, Hellberg C. Macropinocytosis of the PDGF β -receptor promotes fibroblast transformation by H-RasG12V. *Mol Biol Cell*. 2012 Jul;23(13):2571-82.

10: Koumakpayi IH, Le Page C, Delvoye N, Saad F, Mes-Masson AM. Macropinocytosis inhibitors and Arf6 regulate ErbB3 nuclear localization in prostate cancer cells. *Mol Carcinog*. 2011 Nov;50(11):901-12.

11: Lim JP, Wang JT, Kerr MC, Teasdale RD, Gleeson PA. A role for SNX5 in the regulation of macropinocytosis. *BMC Cell Biol*. 2008 Oct 14;9:58.

12: Donepudi M, Resh MD. c-Src trafficking and co-localization with the EGF receptor promotes EGF ligand-independent EGF receptor activation and signaling. *Cell Signal*. 2008 Jul;20(7):1359-67.

Reviewer's Responses to Questions

Experimental quality

Does each figure have the proper controls?

If 'No', please indicate reasons in Comments for Author box below.

Reviewer #1:

- Yes

Reviewer #2:

- Yes

Were the data analyzed using appropriate statistical tests?

If 'No', please indicate reasons in Comments for Author box below.

Reviewer #1:

- Yes

Reviewer #2:

- Yes

Reproducibility

Were experiments performed using adequate number of biological replicates?

If 'No', please indicate reasons in Comments for Author box below.

Reviewer #1:

- Yes

Reviewer #2:

- Yes

Does the methods section provide sufficient detail to permit reproducibility?

If 'No', please indicate reasons in Comments for Author box below.

Reviewer #1:

- Yes

Reviewer #2:

- Yes

Completeness

Are the manuscript's conclusions supported by the data?
If 'No', please indicate reasons in Comments for Author box below.

Reviewer #1:

- Yes

Reviewer #2:

- Yes

Scholarship

Do the authors cite and discuss the merits of data that would argue for and against their conclusion?

If 'No', please indicate reasons in Comments for Author box below.

Reviewer #1:

- Yes

Reviewer #2:

- No

Does the manuscript title & abstract accurately reflect the contents of the manuscript, without hyperbole?

If 'No', please indicate reasons in Comments for Author box below.

Reviewer #1:

- Yes

Reviewer #2:

- Yes

First revision

Author response to reviewers' comments

Reviewer 1: The manuscript by Boyce and colleagues provide novel insights on the route of ATP-driven internalization of PANX1. Authors describe that wild type-PANX1 but not W74A-PANX1 is internalized through macropinocytosis upon ATP stimulation/treatment, and this ATP-driven internalization of PANX1 is inhibited by amiloride pre-treatment. They also utilize pharmacological strategies to target macropinocytosis - EIPA, LY294002 and LatA, and show that pre-treatment of the cells with EIPA, LY294002 or LatA reduces the co-distribution of PANX1 with putative macropinosomes (labelled with 70kDa Dextran). Lastly, they demonstrate that C-terminus of PANX1 interacts with various lipids, most abundantly with PA, PI(4)P, PI(4,5)P2 and PI(3,4,5)P3. Overall, this a well-written manuscript with interesting findings.

Authors improved their manuscript prior to submission to the Biology Open, including their responses to the previous reviewer comments. Minor comment: typo in the name of the PI3K

inhibitor in Figure 4 - LY292004. It should be LY294002. Authors should check this and amend accordingly.

- We thank the reviewer for their enthusiasm for publication. We have amended the typo in Figure 4.

Reviewer 2: This work provides evidence that the PANX1 channel undergoes macropinocytosis from the use of various pharmacological tools. There is evidence that the PANX1 cytoplasmic domain binds to different membrane lipids which may play a role in trafficking but this is ill-defined at present. I have comments that I hope the authors can address:

- We thank the reviewer for their comments and have addressed them individually below.

1) I would like to see some estimate of numbers of PANX1-GFP per cell, and the comparison to physiological levels of PANX1. There is currently also lacking blots for PANX1-GFP and PANX1 in the same cell type which would have been nice to see and compare. The numbers of cell surface 'cargo' molecules is important in considering how many PANX1 molecules are rapidly endocytosed in response to ATP stimulation (of P2X7).

- We appreciate this comment and have added a section in the introduction addressing that overexpression is expected in our system (given both endogenous and ectopic expression in N2a cells). Unfortunately aliquots of the specific stable cell line and appropriate passage numbers used for much of the work presented in this study are no longer available, so we won't be able to provide a blot for comparison between the PANX1-EGFP and endogenous PANX1.

2) In this field, the variety of membrane trafficking portals that govern plasma membrane cargo entry into the endosome-lysosome network is generally classified into clathrin-dependent endocytosis (CDE) and clathrin-independent endocytosis (CIE) e.g. caveolae, macropinocytosis, lipid rafts. For any single PM cargo, it can be envisioned that there is a balance between CDE and CIE at steady-state, with changes depending on environmental conditions and different stimuli. The PANX1 model is no different to say EGFR, transferrin and LDL receptors in this context which all display dynamic associations with different endocytic routes which can be modulated by changing signals and lipid dynamics. The authors need to carefully reframe their language and ideas in this context to make it compatible with these ideas. Please see current references such as:

1: Hemalatha A, Mayor S. Recent advances in clathrin-independent endocytosis. *F1000Res*. 2019 Jan 31;8:F1000 Faculty Rev-138.

2: Sandvig K, Kavaliauskiene S, Skotland T. Clathrin-independent endocytosis: an increasing degree of complexity. *Histochem Cell Biol*. 2018 Aug;150(2):107-118.

3: Cárdenas AM, Marengo FD. Rapid endocytosis and vesicle recycling in neuroendocrine cells. *Cell Mol Neurobiol*. 2010 Nov;30(8):1365-70.

4: Lundmark R, Carlsson SR. Driving membrane curvature in clathrin-dependent and clathrin-independent endocytosis. *Semin Cell Dev Biol*. 2010 Jun;21(4):363-70.

5: Grant BD, Donaldson JG. Pathways and mechanisms of endocytic recycling. *Nat Rev Mol Cell Biol*. 2009 Sep;10(9):597-608.

6: Mayor S, Pagano RE. Pathways of clathrin-independent endocytosis. *Nat Rev Mol Cell Biol*. 2007 Aug;8(8):603-12.

- This has now been addressed specifically in the discussion, where above references were included.

3) It is unclear to me how lipid recognition is linked to endocytosis - are the authors addressing a model specifically here? are such lipids enriched in the macropinocytic structure or are these lipids

localised randomly in the PM? There needs to be some better thinking here as to how lipid turnover is linked to PANX1 recruitment to macropinosomes.

- In the introduction, we describe how membrane lipids later identified as PANX1-CT interactors (ie. PI(4,5)P₂) are involved in macropinocytosis. We have updated the results section to more clearly delineate our logic.

4) What is the balance between PANX1 endocytosis, trafficking, recycling or degradation? All PM cargo undergoes recycling and degradation depending on the nature of the protein, this is unclear here.

- We thank the reviewer for drawing attention to this important point. This has now been specifically addressed in the discussion. Internalized PANX1 in response to extracellular ATP is enriched in recycling endosomes, while at rest, PANX1 is degraded via the endolysosome; however, we do not know the balance of these phenomena in the presence of diverse extracellular stimuli. This will be an exciting topic for future investigation.

5) Other examples of ligand-stimulated macropinocytosis should be referenced:

- 1: Basagiannis D, Zografou S, Murphy C, Fotsis T, Morbidelli L, Ziche M, Bleck C, Mercer J, Christoforidis S. VEGF induces signalling and angiogenesis by directing VEGFR2 internalisation through macropinocytosis. *J Cell Sci.* 2016 Nov 1;129(21):4091-4104.
- 2: Bannach C, Brinkert P, Kühling L, Greune L, Schmidt MA, Schelhaas M. Epidermal Growth Factor Receptor and Abl2 Kinase Regulate Distinct Steps of Human Papillomavirus 16 Endocytosis. *J Virol.* 2020 May 18;94(11):e02143-19.
- 3: Weerasekara VK, Patra KC, Bardeesy N. EGFR Pathway Links Amino Acid Levels and Induction of Macropinocytosis. *Dev Cell.* 2019 Aug 5;50(3):261-263.
- 4: Lee SW, Zhang Y, Jung M, Cruz N, Alas B, Commisso C. EGFR-Pak Signaling Selectively Regulates Glutamine Deprivation-Induced Macropinocytosis. *Dev Cell.* 2019 Aug 5;50(3):381-392.e5.
- 5: Wang HB, Zhang H, Zhang JP, Li Y, Zhao B, Feng GK, Du Y, Xiong D, Zhong Q, Liu WL, Du H, Li MZ, Huang WL, Tsao SW, Hutt-Fletcher L, Zeng YX, Kieff E, Zeng MS. Neuropilin 1 is an entry factor that promotes EBV infection of nasopharyngeal epithelial cells. *Nat Commun.* 2015 Feb 11;6:6240.
- 6: Balaji K, Mooser C, Janson CM, Bliss JM, Hojjat H, Colicelli J. RIN1 orchestrates the activation of RAB5 GTPases and ABL tyrosine kinases to determine the fate of EGFR. *J Cell Sci.* 2012 Dec 1;125(Pt 23):5887-96.
- 7: Sánchez EG, Quintas A, Pérez-Núñez D, Nogal M, Barroso S, Carrascosa ÁL, Revilla Y. African swine fever virus uses macropinocytosis to enter host cells. *PLoS Pathog.* 2012;8(6):e1002754.
- 8: Solis GP, Schrock Y, Hülsbusch N, Wiechers M, Plattner H, Stuermer CA. Reggies/flotillins regulate E-cadherin-mediated cell contact formation by affecting EGFR trafficking. *Mol Biol Cell.* 2012 May;23(10):1812-25.
- 9: Schmees C, Villaseñor R, Zheng W, Ma H, Zerial M, Heldin CH, Hellberg C. Macropinocytosis of the PDGF B-receptor promotes fibroblast transformation by H-RasG12V. *Mol Biol Cell.* 2012 Jul;23(13):2571-82.
- 10: Koumakpayi IH, Le Page C, Delvoye N, Saad F, Mes-Masson AM. Macropinocytosis inhibitors and Arf6 regulate ErbB3 nuclear localization in prostate cancer cells. *Mol Carcinog.* 2011 Nov;50(11):901-12.
- 11: Lim JP, Wang JT, Kerr MC, Teasdale RD, Gleeson PA. A role for SNX5 in the regulation of macropinocytosis. *BMC Cell Biol.* 2008 Oct 14;9:58.
- 12: Donepudi M, Resh MD. c-Src trafficking and co-localization with the EGF receptor promotes EGF ligand-independent EGF receptor activation and signaling. *Cell Signal.* 2008 Jul;20(7):1359-67.

- These studies have been added into the introduction as examples of stimulus-evoked macropinocytosis.

Second decision letter

MS ID#: bio.062272R1

MS Title: ATP increases murine neuroblastoma cell size through a PANX1- and macropinocytosis-dependent mechanism

Authors: Andrew KJ Boyce; Haifei You; Leigh E. Wicki-Stordeur; Leigh Anne Swayne
Article Type: Research Article

I am happy to tell you that your manuscript has been accepted for publication in Biology Open, pending our standard publication integrity checks. It was accepted on 31st October 2025.